# Clinical Predictors for Upper Limb Recovery after Stroke Rehabilitation: Retrospective Cohort Study

**DOI:** 10.3390/healthcare11030335

**Published:** 2023-01-23

**Authors:** Silvia Salvalaggio, Luisa Cacciante, Lorenza Maistrello, Andrea Turolla

**Affiliations:** 1Laboratory of Healthcare Innovation Technology, IRCCS San Camillo Hospital, Via Alberoni 70, 30126 Venice, Italy; 2Padova Neuroscience Center, Università degli Studi di Padova, Via Orus 2/B, 35131 Padova, Italy; 3IRCCS San Camillo Hospital, Via Alberoni 70, 30126 Venice, Italy; 4Department of Biomedical and Neuromotor Sciences–DIBINEM, Alma Mater Studiorum Università di Bologna, Via Massarenti 9, 40138 Bologna, Italy; 5Unit of Occupational Medicine, IRCCS Azienda Ospedaliero-Universitaria di Bologna, Via Pelagio Palagi 9, 40138 Bologna, Italy

**Keywords:** stroke, motor recovery, clinical prediction, upper limb

## Abstract

After stroke, recovery of upper limb (UL) motor function is enhanced by a high dose of rehabilitation and is supposed to be supported by attentive functions. However, their mutual influence during rehabilitation is not well known yet. The aim of this retrospective observational cohort study was to investigate the association between rehabilitation dose and motor and cognitive functions, during UL motor recovery. Inpatients with first unilateral stroke, without time restrictions from onset, and undergoing at least 15 h of rehabilitation were enrolled. Data on dose and modalities of rehabilitation received, together with motor and cognitive outcomes before and after therapy, were collected. Fugl–Meyer values for the Upper Extremity were the primary outcome measure. Logistic regression models were used to detect any associations between UL motor improvement and motor and cognitive-linguistic features at acceptance, regarding dose of rehabilitation received. Thirty-five patients were enrolled and received 80.57 ± 30.1 h of rehabilitation on average. Manual dexterity, level of independence and UL motor function improved after rehabilitation, with no influence of attentive functions on motor recovery. The total amount of rehabilitation delivered was the strongest factor (*p* = 0.031) influencing the recovery of UL motor function after stroke, whereas cognitive-linguistic characteristics were not found to influence UL motor gains.

## 1. Introduction

Stroke is a cerebrovascular disease representing the second cause of death and a major cause of disability worldwide [1]. The most common sequela after stroke is the impairment of Upper Limb (UL) motor function and control, leading to restriction of activities and social participation [2]. Recovery phases after stroke are defined as acute (1–7 days), subacute (7 days–6 months) and chronic (>6 months), with clinical improvement diminishing in accordance with distance from stroke onset, even though sustained by rehabilitation treatments [3]. Nevertheless, recovery is still possible even years after stroke, especially for cognitive domains like language [4,5]. A key factor promoting motor and functional recovery after stroke is dosage of rehabilitation therapy provided. Indeed, trials enrolling patients receiving rehabilitation for a total of 300 h (5 d/wk for 5 h/d), reported clinically relevant improvements of UL function at the Upper Extremity subitem of the Fugl–Meyer Assessment scale (FMA-UE) (i.e., range of score changing from 8 to 11 points) [6]. Recently, a trial aimed to assess maintenance of rehabilitation clinical effects at 6-months follow-up, found that improvements were preserved in patients receiving treatment at least 6 h per day, for three consecutive weeks, even in the chronic phase after stroke [7]. Furthermore, a combination of conventional therapy (CT) and virtual reality (VR) for at least 40 h of rehabilitation was found to enhance clinically relevant improvement in UL motor function, in chronic stroke patients [8]. However, it is not yet known which are the clinical features (e.g., neurological profile; clinical history; level of motor, language, and cognitive functions at baseline) allowing clinicians to predict the recovery potential of a patient before rehabilitation, also considering the treatment pathways followed within the National Health System. Despite some predictors of UL recovery after stroke have been established already (e.g., presence of Motor Evoked Potentials–MEPs, preserved motor function, left lesion site [9]), a recent survey found that 89% of physical therapists (PTs) and occupational therapists (OTs) acknowledge the importance of predicting the potential for recovery after stroke, but only 9% of them actually use prognostic tools in clinical practice [10]. In addition, another under-researched aspect is how cognitive-linguistic and motor functions influence each other and mutually contribute to functional recovery, after stroke. In fact, recent evidence showed that cognitive abilities (especially attention) support motor recovery, throughout large-scale brain networks connecting both cognitive and motor areas [11]. It is therefore reasonable consider these impairments affecting not only the recovery pattern, but also activities of everyday life [12]. Furthermore, cognitive impairments involving memory or executive functions might change responsiveness to motor rehabilitation treatments, affecting the final outcome of targeted interventions after stroke [13].

Another major concern is related to CT contents, indeed, even in studies enrolling patients with severe UL impairments after stroke, less than 30% of PTs and OTs rehabilitation sessions are specifically targeted to arm-related activities [14]. In Europe, PT interventions are generally targeted to body structures and functions with special emphasis on balance and lower limbs training, while occupational therapy (OT) interventions are more targeted to activities of daily living (ADL), domestic and leisure activities, sensory and perceptual training [14]. Recently, a systematic review on the effect of UL-targeted training dosage after stroke found that time spent on specific content of UL-targeted activities was 17% of each PT session, 49% of each OT session, in the acute phase, then ranging widely from 2% to 10% in PT session, and from 23 to 70% in OT session, in the subacute phase [15]. To face this issue, integration of technologies in clinical practice has been improved over the years, allowing to provide high dose of treatment, augmented feedback, and patients’ engagement. Despite these potentials, recommendations to include technologies in current clinical practice are still limited [15].

Despite evidence for factors with positive predictive value for UL recovery (e.g., presence of motor evoked potentials, high level of residual motor function and younger age) being available [16,17], to date, the proper prediction of a patient’s recovery potential induced by rehabilitation treatments is not yet informed by patient clinical characteristics at baseline, neither eventual interactions between cognitive-linguistic and motor functions, nor rehabilitation contents.

Therefore, the study aims to (i) explore clinical features and (ii) potential effect of rehabilitation dose that could influence UL recovery, after stroke.

The rest of the paper is organized as follow: study materials and methods used for the data collection, management and analyses are described. Then, results are presented, and finally the main findings, limitations and suggestions for future research are discussed.

## 2. Materials and Methods

### 2.1. Study Design and Population

This study was a retrospective observational cohort analysis, from data collected on consecutive stroke subjects hospitalized between July 2019 and November 2020 at IRCCS San Camillo Hospital (Venice, Italy). Patients enrolled underwent an initial assessment of motor and cognitive-linguistic functions (T0), whereas only motor functions were reassessed after 20 h of rehabilitation (T1). The original cohort included patients according to following criteria: older than 18 years, diagnosis of a first-ever unilateral cortical-subcortical stroke (ischemic or hemorrhagic) without restriction on time from lesion and with at least 4 weeks of rehabilitation completed. Exclusion criteria were cerebellar or bilateral stroke; unstable medical conditions at time of hospitalization; diagnosis of other neurological and/or psychiatric diseases in addition to stroke (e.g., traumatic brain injury).

The retrospective study design was chosen to analyze data already collected during a standardized screening process at hospital admittance. Therefore, patients hospitalized between July 2019 and November 2020 were contacted by telephone for enrollment and informed on the study purpose, between September and December 2021. Only patients who provided written consent to use their data collected during previous hospitalization were included in the analysis.

For a better reporting of the study, the Strengthening the Reporting of OBservational studies in Epidemiology (STROBE) checklist has been used [18]. The study was conducted in accordance with the Declaration of Helsinki and approved by the Ethics committee of the IRCCS San Camillo hospital (Prot. 2021.20), which is also responsible for the integrity and conduct, the protocol was registered on ClinicalTrials.gov (NCT05478577).

### 2.2. Intervention

During hospitalization lasting 4 weeks at least, patients underwent a motor rehabilitation program consisting at minimum 1 h/day of CT for each day of hospitalization, and one or more hours of other modalities such as UL-specific OT, technology devices (i.e., robotics, virtual reality) for UL and/or lower limb (LL). The treatment program was delivered according to the individual rehabilitation project agreed with the rehabilitation team (e.g., physiotherapist and medical doctor) and tailored on patient’s needs. Each session was adapted to individual clinical condition and ability to perform exercises, accomplishing any harm that may occur (e.g., patients referring shoulder pain, high levels of spasticity). All the technology-based modalities reported are included in the hospital clinical pathways and has been developed and validated through the institutional translational research projects funded by the Italian Ministry of Health and the European Commission.

#### 2.2.1. Technology Devices

Among the therapeutic modalities, technologies for both the upper (UL) and lower (LL) limb were available. Technologies for the UL consisted of Virtual Reality Rehabilitation System (VRRS, Khymeia Group Ltd. Noventa Padovana, Italy), with a computer-based tasks displayed in a virtual scenario. Patients were asked to emulate real arm movements, via a motion tracking system controlling a virtual object [8]. For patients who could benefit from treatments with a robotic device, AMADEO (Tyromotion GmbH, Graz, Austria) was used, an end-effector robot allowing to perform selective voluntary movements of the hand and fingers, controlled by surface electromyography (sEMG) detected from fingers flexors and extensors muscles [19]. Furthermore, among technology devices available, specific UL treatments were delivered by using DIEGO (Tyromotion GmbH, Graz, Austria), an exoskeleton providing arm-weight support while performing virtual tasks, and REMO (Morecognition Ltd. Torino, Italy), a sEMG biofeedback armband for hand movements [20].

Regarding technologies for the LL, the VRRS were used also for LL tasks and balance activities [21]. In addition, the Gait Trainer (GT-I—Reha-Stim, Wisch GmbH & Co., Berlin, Germany), an end-effector robot with body-weight support for walking training was used. Other technologies for LL rehabilitation were the Smart Balance Master (SBM—NeuroCom^®^ Balance Manager, Natus Medical Incorporated, Pleasanton, CA, USA), a semi-immersive balance board providing multisensory balance training exercises with augmented visual biofeedback [22], and the OAK (Khymeia Group Ltd. Noventa Padovana, Italy), an integrated virtual reality system for the assessment and prevention of risk of fall [23]. Finally, Omego (Tyromotion GmbH, Graz, Austria) was available for LL rehabilitation, consisting of a multifunctional robot for pre-walking training (e.g., LL mobilization, muscle strength training, step, press, trunk control) [24].

Each therapy was delivered by a specialized PT for 1 h/day, 5 dd/w, for 3 weeks, with a one-to-one approach. The number of repetitions and type of exercises was chosen by the PT according to clinical judgment and patient’s needs, tailoring difficulties on patient’s ability.

#### 2.2.2. Conventional Therapy

The CT consisted of whole-body exercises selected autonomously by the clinician and performed in a gym or a private room, in a one-to-one setting. Among CT interventions, respiratory therapy was considered. In UL-targeted interventions, patients were asked to perform functional task exercises in each plane including shoulder and elbow flexion-extension, shoulder abduction-adduction, internal-external rotation, circumduction, forearm pronation-supination, both with and without everyday objects. Moreover, exercises were proposed for training coordination, proprioception, and effort resistance capacity in every modality to stimulate patient residual abilities, to reduce compensations and control voluntary muscle activation. If needed, the use of splints or orthosis were considered (e.g., shoulder subluxation, spastic hypertonicity). Each session lasted at least 1 h/day, 5 days/week, for each week of the hospitalization period.

#### 2.2.3. Occupational Therapy

The OT consisted of UL-specific rehabilitation sessions based on the functional use of the limb in ADL (e.g., cooking, dressing, washing), vocational activities (e.g., using a computer, writing), or activities claimed as important by the patient (e.g., sewing). The OT intervention could be delivered in one-to-one, or group settings.

### 2.3. Clinical Data, Assessment and Outcome Measure

Clinical assessments aimed to quantify residuals motor and cognitive-linguistic functions included collection of anamnestic data from digital record of patient medical history, clinical scales measuring the level of functional and sensorimotor capacity of the upper limb, the degree of stroke severity, and communicative-linguistic rating scales.

Demographic and clinical data of each patient were retrieved from digital records of the medical history. Clinical outcomes were retrieved from clinical assessment performed by clinicians (i.e., PT, neuropsychologist, speech language therapist [SLT]). Specifically, data could be tracked back to clinical assessments performed by a PT at the beginning (T0) and end (T1) of a rehabilitation period, and linguistic-cognitive assessments performed by a SLT or neuropsychologist only at T0. The PT and SLT were blinded to rehabilitation intervention, as they were not clinically in charge of the patient. Data on dosage and therapeutic-rehabilitation modalities provided to patients were retrieved from the rehabilitation report filled out by PT.

The primary outcome measure was the FMA-UE, a reliable and validated 66-points outcome measure quantifying arm motor function after stroke [25]. Other clinical outcome measures were: FMA for sensory function (FMA-sensation); Box and Blocks Test (BBT) for gross manual dexterity [26]; Modified Ashworth Scale (MAS) for measuring muscle tone at biceps brachii [27]; Functional Independence Measure (FIM) for autonomy in ADLs [28].

For cognitive and linguistic functions, patients were assessed at baseline with the Oxford Cognitive Screen (OCS), a sensitive screening tool for detection of cognitive deficits after stroke. The scale consists of 10 tasks encompassing five cognitive domains: attention and executive function, language, memory, number processing, and praxis [29].

For each patient, the dose of therapy was quantified both as number of modalities and dose (i.e., total hours of rehabilitation delivered) of intervention received during hospitalization. For the analysis, classes of intervention were defined as follow: total hours of CT (“CT”); total hours of rehabilitation specific for the UL (i.e., UL technologies and OT, “TOT-UL”); total hours of rehabilitation non-specific for the UL (i.e., technologies for LL, “TOT-NUL”); total amount of rehabilitation (i.e., TOT-UL + TOT-NUL + CT = “TOT”). The CT was analyzed only for the primary outcome measure (i.e., FMA-UE).

### 2.4. Sample Size

The sample size of the present study was tailored on the original cohort of stroke patients hospitalized between July 2019 and November 2020 (N = 63) and only those releasing informed consent were finally enrolled and analyzed.

### 2.5. Statistical Analyses

To describe the demographic, clinical and cognitive characteristics of the sample, descriptive statistics (i.e., mean, standard deviation, and percentage) were used. Only a portion of the patients performed the cognitive assessments; therefore, it was decided to perform the descriptive analyses of these variables separately.

Missing data were found to be present for some of the variables. Where the percentage of missing data was less than 25%, the choice was made to impute data using the multivariate imputations by chain equations (MICE) method.

Depending on data distribution, tested through the Shapiro–Wilk test, a paired Student’s *t*-test or Wilcoxon signed rank test was performed to study significant difference in motor outcomes before (T0) and after (T1) rehabilitation. For each outcome measure, effect sizes were calculated by Cohen’s *d* to estimate the standardized effect of rehabilitation [30]. Subsequently, patients were divided in two categories (i.e., Responders, Non-Responders) according to responsiveness to therapy, defined as an improvement greater than the minimally clinically important difference (MCID) or the minimal detectable change (MDC) at clinical outcomes, only if available in the literature. For responsiveness stratification, MCID was considered for FMA-UE (i.e., 5 points), FIM (i.e., 22 points), while MDC for BBT (i.e., 6 points) [31,32,33,34]. To assess whether there was a statistically significant difference in dose of therapy between the Responder and Non-Responder patient groups, Student’s *t* test for unpaired data or Mann–Whitney test for each clinical variable was performed, depending on distribution properties. Because of differences in data completeness, the variables were divided into three groups for models estimation: Clinical Group (i.e., FMA-UE, FMA-sensation, FIM, BBT, MAS-BicBrach, TOT, TOT-UL, TOT-NUL), Cognitive Group (i.e., hearts, recall, shift, assessing attention, memory and executive functions, respectively), and Demographic Group (i.e., Age, Diagnosis, Lesion Side, Time from stroke, Aphasia, Apraxia). Within each group, Generalized Linear Regression Models (GLM) were estimated using the responding variables of each clinical scale as dependent variable and results of other variables in the corresponding groups as independent variables.

Finally, to estimate the overall models of the Responders variable for the primary outcome measure (i.e., FMA-UE), GLM were estimated, using as independent variables the cognitive, demographic, and motor variables found to be significant in the models estimated within the group. For each model, the odds ratios and their 95% confidence intervals (CI) were calculated. In addition, each regression model fitting was assessed by using the following indices [35,36]: (i) McFadden’s index of explained variance (pseudo-R^2^) [37]; (ii) the Scaled Brier Score (sBS), which is a measure of overall accuracy and calculates the average prediction error [38]; (iii) Construction of the Receiver Operating Characteristic (ROC) curve and evaluation of the Area Under the Curve (AUC); and (iv) the Hosmer–Lemeshow test for fit between expected and estimated frequencies ($\chi_{HL}^{2}$; *p*-value) [39].

The regression model fitted the original data if the indices met the following criteria: (i) the more pseudo-R^2^ is close to 1, the more the model is satisfactory; (ii) Brier score for a model can range from 0 (0%) for a perfect model to 1 (100%) for a non-informative model; (iii) an AUC values >0.70 representing a moderately accurate model; (iv) a significant $\chi_{HL}^{2}$ value indicating a bad model fit.

The statistical significance level was set at *p* < 0.05. All the statistical analyses were performed using the free software R Studio 4.0.5 [40].

## 3. Results

Among 63 stroke patients contacted by telephone, 35 of them gave informed consent and were included in the study. Their demographic characteristics (T0) and dose of therapy are described in Table 1.

The UL motor function was moderately impaired before rehabilitation and significantly improved after treatment. Significant improvements were observed also for level of independence and manual dexterity, with effect sizes ranging from low to moderate (Cohen’s *d* < 0.6), as described in Table 2.

The cognitive outcome measures were collected at T0 in those patients needing a cognitive screening (N = 18) and are described in Table 3. Overall, patients tested by OCS presented low-to-moderate cognitive impairments.

After treatment, less than half of the patients improved above the MCID or MDC at the FMA-UE, FIM and BBT, thus classified as responders to therapy (Table 4).

Among the responders to therapy for all the motor outcome measures, the difference on the amount of total dose of rehabilitation was found to be significant only in the FIM group (*p* = 0.031, W = 163.5). Actually, the Non-Responders received more hours of rehabilitation than Responders (Table 5).

Consistently, the Responders and Non-Responders at the FMA-UE, did not receive different doses of rehabilitation (Figure 1.)

Among the Responders at the FMA-UE, the total amount of rehabilitation and a high level of residual independence before rehabilitation (T0) seem to be weakly associated to a higher probability of clinically relevant motor gains. In relation to the cognitive variables assessed before rehabilitation (T0), results showed no significant evidence that attentive functions and independence in ADL influenced motor recovery, positively (Table 6).

## 4. Discussion

The present study explored the association between dose of rehabilitation, cognitive and motor characteristics, in a population of chronic stroke patients undergoing a period of rehabilitation. We observed that the UL motor function (FMA-UE, *p* = 0.005, V = 73), manual dexterity (BBT, *p* = 0.001, V = 9) and level of independence (FIM, *p* = 0.005, V = 88) significantly improved after 80.57 ± 30.1 h of rehabilitation, on average. The overall effect of received intervention was moderate (Cohen’s *d* 0.45 to 0.60). Conversely, sensation functions did not change importantly (FMA-sensation, *p* = 0.501, V = 54.5). Less than half of the patients responded to therapy, according to FMA-UE and FIM (i.e., 34% and 23%, respectively), while almost half of the patients, regarding BBT (i.e., 49%). However, it must be reported that some patients resulted to be non-responders at FMA-UE as their baseline score, higher than 61/66, was within the ceiling effect-zone of the scale.

An utmost finding was that patients classified as non-responders to FIM after treatment, instead received a significant higher dose of rehabilitation, than responders (*p* = 0.031). Conversely, specific interventions for the UL and total dose of rehabilitation specific for the UL did not emerge as significant factors inducing differences between responders and non-responders, confirming that total dose of rehabilitation is more impacting, than dedicated strategies targeted to specific body districts, as previously demonstrated by McCabe et al. [6]. In other words, a higher dose of rehabilitation was delivered to less independent patients (i.e., low FIM score) at hospital acceptance (*p* = 0.031, W = 163.5), therefore to subjects with more severe impairments, thus with larger ranges of improvement expected. It is worth noticed that mild-moderate impairment of muscle tone, sensation, and executive functions at baseline, make patients fully suitable for any potential rehabilitation intervention targeted to the UL, as well as general cognitive functions. Indeed, 12 patients out of the 18 who performed a cognitive screening, presented good levels of attentive, linguistics and mnemonic functions, whereas 13 patients showed good performance of executive functions and no severe cognitive impairment at baseline. Therefore, because of the presence of good cognitive functions in 72% of patients, it was hard to identify the level of cognitive function relevant for empowering improvement of motor function.

Among the responders at FMA-UE, level of independence in ADLs at the beginning of rehabilitation and total dose of intervention accurately predict clinical improvement of UL motor function, as confirmed by the regression model (pseudo-R^2^ = 0.20, AUC = 0.79).

Regarding cognitive variables, the results showed no significant evidence that cognitive-linguistic and attentive functions positively influenced motor recovery, which is not consistent with the present literature [13]. However, it must be reported that according to FMA-UE, the contribution of attentive functions for responding to rehabilitation is close to the significance threshold, even though they seem linked negatively (β = −0.18; *p* = 0.06).

Some limitations of our study need to be acknowledged; the low number of enrolled patients (small sample size) may have underpowered results from the regression models and affected estimation precision, thus confounding potential significant inference. Moreover, the retrospective nature of the study design and the absence of a control group did not allow to explore strong cause-and-effect relationships [41]. Therefore, there is the need to test our findings on larger sample, to improve the model’s statistical fitting and estimation precision for having an accurate view on the potential influence of the cognitive and linguistic functions on motor recovery, more consistent with current literature [11].

## 5. Conclusions

This retrospective cohort study found that total dose is more influential than dose specificity when delivering rehabilitation treatments, for the recovery of motor function, in the chronic phase after stroke. Indeed, higher dose of rehabilitation leads to higher probability of becoming a responder to rehabilitation treatment, for the recovery of the UL motor function. Conversely, the results show that a lower level of independence gain was associated with a higher probability of receiving a larger amount of rehabilitation treatment. Regarding cognitive capability, attentive functions did not seem to be associated with motor recovery, even though their contribution is close to the significance threshold.

In conclusion, the total amount of rehabilitation is confirmed to be the strongest factor contributing to a clinically important improvement in the recovery of UL motor function, after stroke.

To reach firm and strong insights on the predictive factors for motor recovery, improvement of the model’s statistical fitting and estimation precision is required. Therefore, further research should be conducted with longitudinal cohort studies on a larger sample, considering also the enrolment of control cohorts and adjustments for confounding factors.

## Figures and Tables

**Figure 1 healthcare-11-00335-f001:**
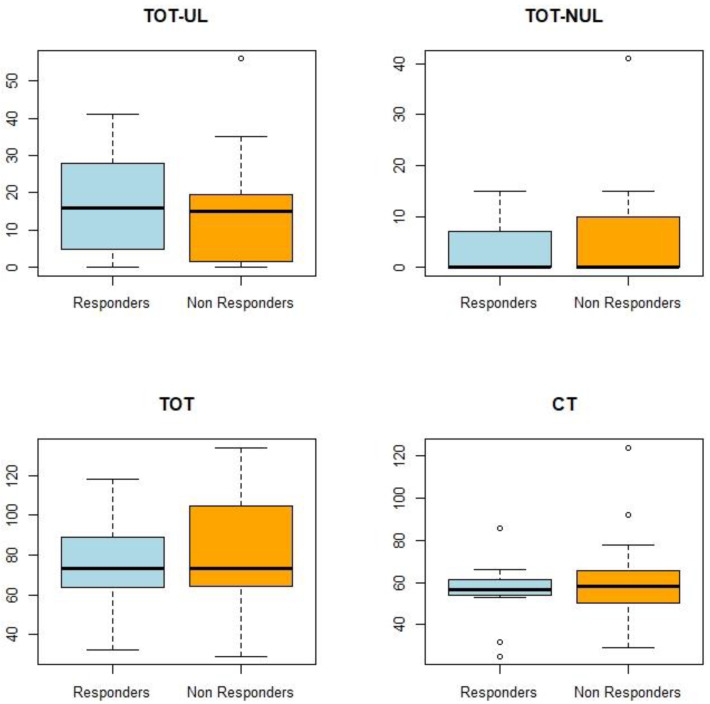
Dose of rehabilitation of the Responders and Non-Responders for FMA-UE. Boxplot of outcomes by Responders and Non-Responders; TOT-UL: total amount of rehabilitation specific for the UL (hours of UL technologies and OT); TOT-NUL: total hours of rehabilitation non-specific for the UL (hours of LL technologies); TOT: total amount of rehabilitation (hours); CT: conventional therapy.

**Table 1 healthcare-11-00335-t001:** Demographic characteristics at baseline (T0) and dose of therapy.

Patients (N = 35)	
Age, years, mean ± SD	65.26 ± 16.2
Diagnosis, ischemic/hemorrhagic, *n* (%)	25 (71%)/10 (29%)
Lesion Side, right/left, *n* (%)	24 (69%)/11 (31%)
Time from stroke, months, mean ± SD	26.72 ± 67.1
Aphasia, yes/no, *n* (%)	14 (40%)/20 (60%)
Apraxia, yes/no, *n* (%)	2 (6%)/31 (94%)
TOT, mean ± SD	80.57 ± 30.1
TOT-UL, mean ± SD	13.4 ± 14.19
TOT-NUL, mean ± SD	5.34 ± 9.5
CT, mean ± SD	64.03 ± 23.46

Values are expressed as mean ± standard deviation (SD) for quantitative measures, and frequency (*n*) and percentage (%) for discrete variables; N: number of patients; TOT: total amount of rehabilitation (hours); TOT-UL total amount of rehabilitation specific for the UL (hours of UL technologies and OT); TOT-NUL: total hours of rehabilitation non-specific for the UL (hours of LL technologies); CT total hours of conventional therapy of the TOT.

**Table 2 healthcare-11-00335-t002:** Motor outcome measures before (T0) and after (T1) rehabilitation.

Outcome Measure (N = 35)	T0		T1		Within Group *p*-Value	Effect Size (Cohen’s *d*)
	Mean ± SD	Median (IQR)	Mean ± SD	Median (IQR)		
FMA-UE	31.60 ± 24.4	34 (46.5)	37.20 ± 23.2	45 (45)	0.005 *	0.45
FMA-sens	18.29 ± 7.3	22 (12)	19.11 ± 6.1	23 (11.5)	0.501	0.15
FIM	86.17 ± 29.7	88 (58)	97.69 ± 26.8	109 (40)	0.005 *	0.6
BBT	16.60 ± 17.7	14 (32)	24.63 ± 20.5	29 (43)	< 0.001 *	0.59
MAS-BicBrach	0.91 ± 0.9	1 (2)				

Values are expressed as mean ± standard deviation (SD) and Median and Interquartile range (IQR). FMA-UE: Fugl–Meyer Assessment Upper Extremity; FMA-sens: Fugl–Meyer Assessment–sensation; FIM: Functional Independence Measure; BBT: Box and Blocks Test; MAS-BicBrach: Modified Ashworth Scale at Biceps Brachii muscle. Wilcoxon singed-rank test was used for within analyses. Significance was established at *p* < 0.05 *.

**Table 3 healthcare-11-00335-t003:** Oxford Cognitive Scale (OCS) evaluated before (T0) rehabilitation.

Outcome Measure (N = 18)	T0 Mean ± SD
Hearts	44.83 ± 6.5
Recall	2.78 ± 1.2
Shift	1.72 ± 4

Values are expressed as mean ± standard deviation (SD). N: number of patients. Hearts: attentive function; Recall: memory; Shift: executive functions.

**Table 4 healthcare-11-00335-t004:** Patients responding to therapy in the motor domain.

Outcome Measure (N = 35)	Responders/Non-Responders *n* (%)
FMA-UE	12 (34%)/23 (66%)
FIM	8 (23%)/27 (77%)
BBT	17 (49%)/18 (51%)

Values are expressed as frequency (n) and percentage (%). FMA-UE: Fugl–Meyer Assessment Upper Extremity; FIM: Functional Independence Measure; BBT: Box and Blocks Test.

**Table 5 healthcare-11-00335-t005:** Comparison between dose (hours) of rehabilitation between Responders and Non-Responders for UL motor function.

Dose for Each Outcome Measure	Responders		Non-Responder		Between Groups
	Mean ± SD	Median (IQR)	Mean ± SD	Median (IQR)	
**FMA-UE**	*n* = 12		*n* = 23		*n* = 23
TOT-UL	17.17 ± 14.06	16 (18.5)	11.43 ± 14.16	15 (17)	*p* = 0.607
TOT-NUL	3.67 ± 6.64	0 (3.5)	6.22 ± 10.77	0 (10)	*p* = 0.221
TOT	76.33 ± 22.71	73.5 (21.25)	82.78 ± 33.55	72 (40.5)	*p* = 0.524
CT	72.5 ± 33.7	56.5 (26)	56.26 ± 12.17	58 (13.5)	*p* = 0.300
**FIM**	N = 8		N = 27		
TOT-UL	12.00 ±12.68	10.5 (19.25)	13.82 ± 14.81	14 (20)	*p* = 0.841
TOT-NUL	1.88 ± 5.30	0 (0)	6.37 ± 10.32	0 (12)	*p* = 0.193
TOT	61.25 ± 14.96	63.5 (13)	86.29 ± 31.21	75 (44)	*p* = 0.031 *
**BBT**	N = 17		N = 18		
TOT-UL	12.29 ± 15.79	6.0 (20)	14.44 ± 12.88	15.5 (19)	*p* = 0.511
TOT-NUL	4.94 ± 8.33	0 (8)	5.72 ± 10.77	0 (11.25)	*p* = 0.934
TOT	82.94 ± 38.34	70 (53)	78.33 ± 20.37	74 (23.25)	*p* = 0.591

Values are expressed as mean ± standard deviation (SD) and Median and interquartile range (IQR). * *p* values < 0.05; Mann–Whitney test was used for between analysis. FMA-UE: Fugl–Meyer Assessment Upper Extremity; FIM: Functional Independence Measure; BBT: Box and Blocks Test; TOT-UL: total amount of rehabilitation specific for the UL (hours of UL technologies and OT); TOT-NUL: total hours of rehabilitation non-specific for the UL (hours of LL technologies); CT: conventional therapy (hours); TOT: total amount of rehabilitation (hours).

**Table 6 healthcare-11-00335-t006:** Relationship between the FMA-UE Responders and clinical and rehabilitation features.

Regression Model	β ± SE	Pseudo-R2	sBS	AUC	PHL
Intercept FIM TOT	0.06 ± 1.66 −0.03 ± 0.02 0.02 ± 0.02	0.20	0.26	0.79	*p* = 0.33
Intercept Heart * (*p* = 0.06)	7.34 ± 4.25 −0.18 ± 0.09	0.18	0.24	0.70	*p* = 0.47
Intercept TOT * (*p* = 0.09) Hearts	7.06 ± 4.8 0.04 ± 0.02 −0.25 ± 0.12	0.36	0.42	0.87	*p* = 0.24

The outcomes are displayed with: Estimate of regression coefficient with Standard Error (β ± SE); McFadden’s index of explained variance (pseudo-R2); Scaled Brier Score (sBS); Area Under the Curve (AUC); *p*-value of the Hosmer–Lemeshow test (PHL). Significance was established at *p* < 0.05 *.

## Data Availability

The datasets that support the findings of this study are available from the corresponding author upon request.

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
