# Peer review of "Clinical Predictors for Upper Limb Recovery after Stroke Rehabilitation: Retrospective Cohort Study"

_healthcare, 2023, doi:10.3390/healthcare11030335_

Round 1

Reviewer 1 Report

In this study, the authors conducted a retrospective observational cohort study on stroke patients. They used logistic regression models to detect any associations between Upper Limb motor improvement and motor and cognitive-linguistic features at acceptance, regardless of rehabilitation dose received. They enrolled thirty-five patients who received an average of 80.57 30.1 hours of rehabilitation, resulting in improvements in manual dexterity,

independence, and Upper Limb motor function. The authors found that attentive functions had no effect on motor recovery. The total amount of rehabilitation has been shown to be the most influential factor in the recovery of Upper Limb motor function after stroke.

My observations are as follows:

1.     This manuscript is seriously lacking in a thorough literature review.

2.     Please include/discuss at least 20 recent studies in this area, compare your study to those, and inform the readers about the new developments that you have made in your work.

3.     The term "retrospective cohort study" must be defined, and its application, purpose, and procedure in this study must be justified and explained.

4.     The study's limitations must be specified.

5.     There are no suggestions for future research.

6.     The sample size of 35 is insufficient. Add some references to back up small sample sizes.

7.     If a questionnaire was distributed to the patients, it may be included as an appendix.

8.     The study's findings must be clearly highlighted in the conclusion section.

9.     Try including some graphics (charts) in your research.

10.  There are also formatting and language issues. Please look into it.

11.  At the end of the Introduction section, include a paragraph on manuscript organisation.

Reviewer 2 Report

Dear Authors, the manuscript is of particular interest. I have the following concerns:

I would suggest more structure to the abstract. In fact, the objective should refer to the form "this (...) study aimed to". The methodological section is completely missing, with an outline of the population (its eligibility), intervention and above all clear outcome measures. Especially on line 27, I would recommend entering an objective data with significance level.

Line 33, I would start from the description of the disorder. the concept of chronic, the rehabilitation need and the facets of the disease with regard to the upper limb. ref:

https://pubmed.ncbi.nlm.nih.gov/28734936/

https://pubmed.ncbi.nlm.nih.gov/34757009/

https://pubmed.ncbi.nlm.nih.gov/29453980/

Throughout the background of the introduction I would recommend adding references

References in square brackets should be inserted before the period.

Line 70 I would also develop these manuscripts:

https://pubmed.ncbi.nlm.nih.gov/22023891/

https://pubmed.ncbi.nlm.nih.gov/31925838/

Clearly the originality of your manuscript should be emphasized above all in that very broad and difficult to study concept of "sub-acute" stroke (from what I ascertain on line 82-83, this eligibility should be inserted in the abstract to guide readers).

Line 80, have you used any validated functional score cut-offs?

95 Exposure? please explain. I would suggest more detail, especially regarding the use of robotic therapy and VR, what kind.. for how long.. This is fundamental to understand the impact and role of the approach compared to conventional therapy

104 Bibliographic references are totally missing

2.2.3. Technology devices: the entire paragraph should be placed before (conventional therapy) with greater attention to detail, without listing only the devices.

  151 selected time points I recommend that they be described in the design

172-176 Expose these analyzes before during the interventions.

Table 5. Mann Whitney U would not be needed?

278 are they really chronic?

279 put the references to the scales used..

288 “Non-responders to FIM” is not clear, it seems that patients do not respond to the rating scale; not ingredients, but approaches

Separate and expand the limitations section, an evaluation of gender is missing, among other things the presence or absence of a neglect, moreover the age of participants had a 16 SD. the nature of the design may not allow definitive conclusions, a follow up on a “true” chronic stroke is missing.. it is longitudinal but with a small time window..

316 chronic?

On the other hand, as previously mentioned Coupar et al. states that “The most important predictive factors for upper limb recovery following stroke appears to the initial severity of motor impairment or function.”

Reviewer 3 Report

This manuscript presents a statistical analysis on the effects of rehabilitation dosage and type into the recovery of patients. The main finding is that it is the amount of rehabilitation and not its specificity what is relevant for recovery and for prediction of outcomes. The are a number of comments to put on this article:

1) As the authors recognize, the sample is rather small for regression, and it is moderate but valid for a sound statistical analysis. Thus, the main conclusions need further and stronger evidence to be accepted.

2) The presentation of results in this study is quite succinct: the main outcomes appear in Tables 2 and 5, while the remaining tables are basically descriptive statistics (Table 6 shows no influence of the attentive factors, though). However, those results identify the total hours of rehabilitation under the metric FIM as the only statistically significant correlation. This is not exactly a predictor or a model, as the paper title suggests.

3) Some brief explanation of the purpose and meaning of each statistical test used in the study should be provided in the text.

Round 2

Reviewer 1 Report

Please consider my previous comments and update your article accordingly. This version contains no significant changes in my opinion. On “paper organization”, I mean a paragraph about paper organisation that begins, "the rest of the paper is organised.......",

Only clean copy with changed sentences highlighted in yellow should be submitted.

Reviewer 2 Report

Again.. The methodological section (in the abstract) completely missing, with a profile of the population (its eligibility, post-acute, sub-acute or chronic?), interventions (what did you administer methodologically) and above all clear outcome measures (how did you measure the interventions previously described)…

As in the previous review, the manuscript can give insights to the readers, but it must necessarily respect a strict methodology. it is necessary, indeed fundamental, to describe in the methods so it will be presented in the results

Population the eligibility of the stroke phase cannot be ignored.

“The Stroke Roundtable Consortium proposed to designate the first 24 h as the hyperacute phase, the first 7 days as the acute phase, the first 3 months as the early sub-acute phase, the months 4–6 as the late sub-acute phase, and from 6 months on as the chronic phase” ref: https://pubmed.ncbi.nlm.nih.gov/33324923/

Is the baseline a subacute stroke, after 1 month of rehabilitation treatment (among other things, what type of treatment)? Or were Chronics entered (6 months after the "first unilateral" event or even years later?), who potentially have several months of therapy?

The eligibility is too large...

L142 The checklist does not mention “exposure”, I recommend removing and following the guidelines as stated by the authors

L149 (adjust abbreviations) CT? conventional? What do you mean by conventional? How many hours, what treatment models? OT, what do you mean? What kind of robotic device? When is it included in the rehabilitation framework?

Is the study goal still the appropriate rehabilitation dose? No dose is defined in the methods

As in the previous review, I wanted you to provide a clear “intervention” paragraph, not a device shopping list.. ending the section with 1hour/day, 5 daysd/week, for 3 weeks.

How much VRSS, how much AMADEO, how much DIEGO, how much REMO, how much GAIT trainer, etc.. ????

I read at least 10 devices..

No mention of the objective of "potential effect of rehabilitation dose that could influence UL"

Time from stroke, months, mean ± SD 26.72 ± 67.1 … 2 years after the event, but with a huge standard deviation

The results are not understood because methodologically it is not clear how they will be reported. In addition, the figures are not scientific, there is no SD for bar graphs and being inhomogeneous groups, with non-normal distribution, boxes and whiskers should be presented.
